https://doi.org/10.1038/s41467-021-21237-w | **OPEN**

# SARS-CoV-2 antibody prevalence in England following the first peak of the pandemic

Helen Ward [1,2,3✉], Christina Atchison[1], Matthew Whitaker[1], Kylie E. C. Ainslie [1,4], Joshua Elliott[1], Lucy Okell [1,4], Rozlyn Redd[1], Deborah Ashby[1], Christl A. Donnelly [1,4,5], Wendy Barclay[2,6], Ara Darzi [2,7], Graham Cooke [2,6], Steven Riley[1,4] & Paul Elliott [1,2,8✉]

England has experienced a large outbreak of SARS-CoV-2, disproportionately affecting people from disadvantaged and ethnic minority communities. It is unclear how much of this excess is due to differences in exposure associated with structural inequalities. Here, we report from the REal-time Assessment of Community Transmission-2 (REACT-2) national study of over 100,000 people. After adjusting for test characteristics and re-weighting to the population, overall antibody prevalence is 6.0% (95% CI: 5.8-6.1). An estimated 3.4 million people had developed antibodies to SARS-CoV-2 by mid-July 2020. Prevalence is two- to three-fold higher among health and care workers compared with non-essential workers, and in people of Black or South Asian than white ethnicity, while age- and sex-specific infection fatality ratios are similar across ethnicities. Our results indicate that higher hospitalisation and mortality from COVID-19 in minority ethnic groups may reflect higher rates of infection rather than differential experience of disease or care.

[1] School of Public Health, Imperial College London, London, UK. [2] National Institute for Health Research Imperial Biomedical Research Centre, London, UK. [3] Imperial College Healthcare NHS Trust, London, UK. [4] MRC Centre for Global Infectious Disease Analysis Imperial College London, London, UK. [5] Department of Statistics, University of Oxford, Oxford, UK. [6] Department of Infectious Disease, Imperial College London, London, UK. [7] Institute of Global Health Innovation, Imperial College London, London, UK. [8] MRC Centre for Environment and Health, Imperial College London, London, UK. ✉email: h.ward@imperial.ac.uk; p.elliott@imperial.ac.uk

England has experienced a large outbreak of SARS-CoV-2 infection leading to the highest excess mortality in Europe by June 2020[1]. The first recorded COVID-19 death occurred on 28 February, with in-hospital deaths peaking by mid-April[2]. Hospital admission and mortality data show an asymmetrical burden of COVID-19 in England, with high rates in older people and those living in long-term care, and in people of minority ethnic groups, particularly Black and Asian (mainly South Asian) individuals[3–6]. It is unclear how much of this excess is due to differences in exposure to the virus, e.g. related to workplace exposures and structural inequality, and how much is due to differences in outcome, including access to health care[7–9].

As part of the UK Government's response to controlling the spread of the virus, on March 23 it announced a national lockdown that prohibited all but essential activities. The UK came out of lockdown from mid-May as restrictions were gradually eased as more business were allowed to reopen and the public was encouraged to use face coverings in situations when social distancing could not be maintained.

Antibody data provide a long-lasting measure of SARS-CoV-2 infection, enabling analyses of the timing and extent of the recent epidemic. Most infected people mount an IgG antibody response detectable after 14–21 days although levels may start to wane after ~90 days[10]. Uncertain validity of the available antibody tests, inconsistencies in sampling methods, small numbers and use of selected groups have made many studies difficult to interpret[11]. Different acceptability criteria may apply to community-based studies where population-wide results are required than for studies focused on individual risk[11–14]. While not generally approved for individual care, self-administered lateral flow immunoassay (LFIA) tests done at home provide a means for obtaining reliable community-wide prevalence estimates rapidly and at scale, at reasonable cost[15,16], by adjusting the results for known test performance[17].

Here, we obtained estimates of the cumulative community prevalence of IgG antibodies for SARS-CoV-2 infection among a representative sample of over 100,000 adults aged over 18 years in England, and specific sub-groups of the population, e.g. by ethnicity and occupation, to mid-July 2020[18]. We used home-based self-testing with a LFIA that had been extensively evaluated for sensitivity and specificity in both laboratory and clinic settings and for acceptability and usability among the public[19,20]. The tests were delivered by post to randomly selected individuals who were given detailed instructions (including by video) on how to carry out the procedure. Participants were asked to upload a photograph of the completed test and to complete a brief questionnaire either online or by telephone (see the "Methods" section and published protocol[18]). As well as measuring community prevalence and identifying groups at most risk of infection, we estimated the total number of infected individuals in England and the infection fatality ratio (IFR) overall and by age, sex and ethnicity.

## Results

Of the 121,976 people who were sent test kits, 109,076 (89.4%) completed the questionnaire of whom 105,651 (96.9%) completed the test, during the period 20 June–13 July 2020; 5544 (5.2%) were IgG positive, 94,364 (89.3%) IgG negative and 5743 (5.4%) reported an invalid or unreadable result, giving a crude prevalence of 5.6% (95% CI 5.4–5.7). After adjusting for the performance characteristics of the test and re-weighting to be representative of the population, overall antibody prevalence was 6.0% (95% CI: 5.8–6.1). This equates to 3.36 (3.22, 3.51) million adults in England who had antibodies to SARS-CoV-2 in England to mid-July 2020.

Prevalence was highest at ages 18–24 years (7.9%, 95% CI 7.3, 8.5) and in London (13.0%, 95% CI 12.3, 13.60) (Supplementary Table 1). Highest prevalence by ethnic group was found in people of Black (includes Black Caribbean, African and Black British) (17.3%, 95% CI 15.8, 19.1) and Asian (mainly South Asian) ethnicities (11.9%, 95% CI 11.0, 12.8), compared to 5.0% (95% CI 4.8, 5.2) in people of white ethnicity (Supplementary Table 1). There was some variation within these broad ethnic categories (Supplementary Table 2), with the highest prevalence in people of Black African ethncity (19.21%, 95% CI 15.57, 23.42). The increased prevalence among non-white ethnicities was partially but not fully explained by covariates. For example, in an unadjusted logistic regression model, compared to white ethnicity, Black ethnicity was associated with a three-fold increase in odds of being antibody positive (OR 3.2, 95% CI 2.7, 3.9) which reduced to OR 2.0 (1.7, 2.5) after adjustment for covariates (Table 1, Supplementary Fig. 1). Essential workers, particularly those with public-facing roles, also had increased prevalence: among those working in residential care facilities (care homes) with client-facing roles, prevalence was 16.5% (95% CI 13.7, 19.8) and it was 11.7% (95% CI 10.5–13.1) among health care workers with patient contact, with 3-fold (3.1; 2.5, 3.8) and 2-fold (2.1; 1.9, 2.4) odds of infection, respectively, compared with non-essential workers (Table 1, Supplementary Table 3, Supplementary Fig. 1). Those living in more deprived areas or in larger households, particularly without children, had higher prevalence than those in more affluent areas or who lived alone, although the increased odds were partially attenuated in the adjusted models. In contrast, higher household income was associated with increased prevalence of antibodies (Table 2, Supplementary Fig. 1), which was greater in younger age groups (Supplementary Table 4).

Of the 5544 IgG positive people, 3406 (61.4%; 60.1, 62.7) reported one or more typical symptoms (fever, persistent cough, loss of taste or smell), 353 (6.4%; 5.8, 7.0) reported atypical symptoms only, and 1785 (32.2%; 31.0, 33.4) reported no symptoms. This varied by age, with people over 65 being more likely to report no symptoms (392/801, 48.9%, 45.4, 52.4) than those aged 18–34 (418/1,393, 30.0%, 27.6, 32.4) or 35–64 years (975/3,350, 29.1%, 27.6, 30.6) ($P < 0.001$). Prevalence was higher in those with more severe symptoms, and who had contact with a confirmed or suspected case. Those who were overweight or obese had higher prevalence than those with normal weight, while current smokers had a lower prevelance than non-smokers consistent with findings from other studies[4,21] (3.2% vs. 5.2%, OR 0.6, 95% CI 0.6, 0.7) (Table 1, Supplementaray Fig. 1, Supplementary Table 3).

Figure 1 shows how the epidemic evolved between January and June 2020. An epidemic curve was generated from dates of reported suspected or confirmed COVID-19 among symptomatic cases with antibodies ($n = 3493$; asymptomatic individuals and symptomatic people whose date of infection was unknown are excluded). The plot shows the epidemic curve from the present study, alongside national mortality for England by date of death: this tracks 2–3 weeks later than our epidemic curve, which peaked in the first week of April at the height of the epidemic in England. Figure 2 shows the proportionate distribution of cases from our data by employment. As the epidemic grew there was a shift towards a greater proportion of cases in essential workers, particularly those in resident-facing and patient-facing roles in care homes and health care.

The estimated community IFR (excluding care homes) was 0.90% (0.86, 0.94). It was higher in males (1.07%, 1.00, 1.15) than females (0.71%, 0.67, 0.75) and increased with age from 0.52% (0.49, 0.55) at ages 45–64 years to 11.64% (9.22, 14.06) at ages 75+ years (Table 2). Sensitivity analyses indicate an IFR as high as

**Table 1 Logistic regression analysis for prevalence of IgG antibodies to SARS-CoV-2.**

| Categories | Unadjusted | Adj: age, sex, region | Adj: age, sex, ethnicity, region, IMD quintile, household size, employment, child in house (y/n) |
|---|---|---|---|
| *Sex* | | | |
| Male | Reference | Reference | Reference |
| Female | 1.02 [0.97,1.08] | 1.00 [0.95,1.06] | 0.97 [0.91,1.02] |
| *Age* | | | |
| 18–24 | 1.28 [1.14,1.44] | 1.31 [1.17,1.47] | 1.25 [1.11,1.42] |
| 25–34 | 1.25 [1.14,1.37] | 1.22 [1.11,1.34] | 1.18 [1.07,1.30] |
| 35–44 | Reference | Reference | Reference |
| 45–54 | 1.08 [0.99,1.18] | 1.13 [1.03,1.23] | 1.11 [1.02,1.22] |
| 55–64 | 0.98 [0.90,1.07] | 1.04 [0.95,1.13] | 1.07 [0.97,1.18] |
| 65–74 | 0.63 [0.57,0.70] | 0.68 [0.61,0.76] | 0.85 [0.75,0.97] |
| 75+ | 0.63 [0.54,0.73] | 0.68 [0.59,0.79] | 0.89 [0.76,1.06] |
| *Ethnicity* | | | |
| White | Reference | Reference | Reference |
| Mixed | 1.56 [1.27,1.90] | 1.21 [0.99,1.48] | 1.24 [1.01,1.52] |
| Asian | 2.04 [1.83,2.28] | 1.56 [1.39,1.75] | 1.44 [1.28,1.62] |
| Black | 3.21 [2.67,3.87] | 2.27 [1.88,2.75] | 2.02 [1.67,2.46] |
| Other | 2.11 [1.67,2.66] | 1.50 [1.18,1.90] | 1.49 [1.17,1.89] |
| *Region* | | | |
| North East | 1.22 [1.05,1.43] | 1.21 [1.04,1.42] | 1.20 [1.03,1.41] |
| North West | 1.33 [1.21,1.47] | 1.33 [1.20,1.47] | 1.33 [1.20,1.47] |
| Yorkshire and The Humber | 0.96 [0.84,1.10] | 0.96 [0.84,1.10] | 0.97 [0.85,1.11] |
| East Midlands | 1.05 [0.95,1.16] | 1.04 [0.94,1.16] | 1.05 [0.94,1.16] |
| West Midlands | 1.27 [1.14,1.42] | 1.27 [1.14,1.42] | 1.28 [1.15,1.42] |
| East of England | 1.25 [1.13,1.37] | 1.24 [1.13,1.36] | 1.25 [1.14,1.38] |
| London | 2.59 [2.37,2.84] | 2.48 [2.27,2.72] | 2.34 [2.13,2.57] |
| South East | Reference | Reference | Reference |
| South West | 0.82 [0.72,0.93] | 0.83 [0.73,0.94] | 0.84 [0.74,0.95] |
| *IMD quintile* | | | |
| Most deprived: 1 | 1.32 [1.20,1.45] | 1.15 [1.04,1.26] | 1.11 [1.00,1.22] |
| 2 | 1.14 [1.05,1.24] | 0.97 [0.89,1.06] | 0.94 [0.86,1.03] |
| 3 | 1.07 [0.99,1.16] | 1.00 [0.93,1.09] | 0.99 [0.91,1.07] |
| 4 | 0.99 [0.92,1.07] | 0.96 [0.88,1.03] | 0.95 [0.88,1.03] |
| Least deprived: 5 | Reference | Reference | Reference |
| *Highest educational level reached* | | | |
| No qualification | Reference | Reference | Reference |
| Other | 1.24 [1.07,1.43] | 1.13 [0.98,1.30] | 1.09 [0.94,1.26] |
| GCSE | 1.11 [0.99,1.25] | 0.94 [0.83,1.05] | 0.94 [0.83,1.06] |
| Post-GCSE qualification | 1.20 [1.07,1.34] | 0.97 [0.86,1.09] | 0.94 [0.84,1.06] |
| Degree or above | 1.34 [1.21,1.50] | 1.00 [0.89,1.12] | 0.96 [0.85,1.08] |
| *Gross household income* | | | |
| £0–14,999 | 0.86 [0.76,0.97] | 0.86 [0.76,0.98] | 0.87 [0.77,1.00] |
| £15,000–49,999 | Reference | Reference | Reference |
| £50,000–149,999 | 1.19 [1.11,1.28] | 1.06 [0.99,1.14] | 1.08 [1.00,1.16] |
| >£150,000 | 1.73 [1.52,1.97] | 1.35 [1.17,1.54] | 1.39 [1.21,1.61] |
| *Employment* | | | |
| Healthcare (patient-facing) | 2.04 [1.82,2.29] | 2.19 [1.95,2.46] | 2.09 [1.86,2.35] |
| Healthcare (other) | 1.10 [0.87,1.40] | 1.15 [0.90,1.46] | 1.11 [0.87,1.42] |
| Care home (client-facing) | 2.90 [2.37,3.55] | 3.19 [2.60,3.92] | 3.11 [2.53,3.83] |
| Care home (other) | 1.46 [0.81,2.64] | 1.67 [0.92,3.02] | 1.66 [0.92,3.01] |
| Other essential worker[a] | 1.05 [0.98,1.13] | 1.11 [1.03,1.20] | 1.10 [1.02,1.19] |
| Other worker[b] | Reference | Reference | Reference |
| Not in employment | 0.72 [0.67,0.77] | 0.86 [0.80,0.93] | 0.84 [0.78,0.91] |
| *Household size* | | | |
| 1 | Reference | Reference | Reference |
| 2 | 1.03 [0.94,1.12] | 1.04 [0.95,1.14] | 1.07 [0.98,1.17] |
| 3 | 1.24 [1.13,1.37] | 1.10 [1.00,1.22] | 1.19 [1.08,1.33] |
| 4 | 1.30 [1.18,1.43] | 1.15 [1.04,1.27] | 1.29 [1.16,1.44] |
| 5 | 1.49 [1.32,1.69] | 1.30 [1.14,1.47] | 1.47 [1.27,1.68] |
| 6 | 1.78 [1.49,2.14] | 1.51 [1.26,1.82] | 1.66 [1.36,2.03] |
| 7+ | 2.10 [1.65,2.68] | 1.72 [1.34,2.21] | 1.85 [1.42,2.41] |
| *One or more children (under 18) in household* | | | |
| No | Reference | Reference | Reference |
| Yes | 1.09 [1.03,1.15] | 0.96 [0.90,1.03] | 0.81 [0.75,0.88] |

**Table 1 (continued)**

| Categories | Unadjusted | Adj: age, sex, region | Adj: age, sex, ethnicity, region, IMD quintile, household size, employment, child in house (y/n) |
|---|---|---|---|
| *History of COVID-19 symptoms* | | | |
| No symptoms | Reference | Reference | Reference |
| Mild symptoms | 1.39 [1.10,1.76] | 1.47 [1.16,1.86] | 1.57 [1.23,2.00] |
| Moderate symptoms | 1.71 [1.36,2.15] | 1.84 [1.46,2.32] | 2.04 [1.61,2.59] |
| Severe symptoms | 2.31 [1.82,2.93] | 2.51 [1.97,3.19] | 2.75 [2.15,3.52] |
| *BMI* | | | |
| Underweight (<18.5) | 0.99 [0.77,1.28] | 0.89 [0.69,1.15] | 0.88 [0.68,1.14] |
| Normal (18.5–24.9) | Reference | Reference | Reference |
| Overweight (25–29.9) | 1.09 [1.02,1.16] | 1.18 [1.10,1.26] | 1.17 [1.09,1.25] |
| Obese (≥30) | 1.11 [1.03,1.20] | 1.19 [1.10,1.29] | 1.18 [1.09,1.28] |
| *Current smoker* | | | |
| Yes | 0.70 [0.63,0.77] | 0.65 [0.58,0.71] | 0.64 [0.58,0.71] |
| No | Reference | Reference | Reference |

Unadjusted and adjusted odds ratios [95% confidence intervals]. Forest plot shown in Supplementary Fig. 1. Unadjusted odds ratios were obtained from univariable logistic regression for the covariate of interest. Adjusted odds ratio were obtained by performing multivariable logistic regression for the covariate of interest with age, sex, and region (column 3); and age, sex, ethnicity, region, IMD quintile, education, gross household income, employment, household size, and child in house (y/n) (column 4) [a]Essential worker: List of essential workers as defined by the UK Government https://www.gov.uk/guidance/coronavirus-covid-19-getting-tested#essential-workers; [b]Other worker: Worker not working in health or social care or on the UK Government list of essential workers. Source data are provided with this paper.

**Table 2 Infection fatality ratio (IFR) and numbers of SARS-CoV-2 infections by age, sex, ethnicity.**

| Category | Population size (000s) | SARS-CoV-2 antibody prevalence % [95% CI][a] | Confirmed COVID-19 deaths | Infection fatality ratio % [95% CI][b] | Estimated number of infections (000s) [95% CI] |
|---|---|---|---|---|---|
| (a) IFR and total infections overall and by age and sex (excluding care home deaths) | | | | | |
| Total | 56,287 | 5.96 [5.70, 6.75] | 30,180 | 0.90 [0.86, 0.94] | 3362 [3217; 3507] |
| *Sex* | | | | | |
| Male | 27,828 | 6.17 [5.76, 6.59] | 18,575 | 1.07 [1.00, 1.15] | 1730 [1615; 1845] |
| Female | 28,459 | 5.75 [5.42, 6.09] | 11,600 | 0.71 [0.67, 0.75] | 1634 [1540; 1728] |
| *Age* | | | | | |
| 15–44 | 21,335 | 7.20 [6.73, 7.66] | 524 | 0.03 [0.03, 0.04] | 1536 [1437; 1635] |
| 45–64 | 14,406 | 6.18 [5.78, 6.58] | 4657 | 0.52 [0.49, 0.55] | 895 [837; 953] |
| 65–74 | 5576 | 3.16 [2.67, 3.66] | 5663 | 3.13 [2.65, 3.61] | 181 [153; 209] |
| 75+ | 4778 | 3.30 [2.53, 4.08] | 19,330 | 11.64 [9.22, 14.06] | 166 [131; 201] |
| (b) IFR and total infections by ethnicity, age and sex (including care home deaths[c]) | | | | | |
| *Female <65* | | | | | |
| White | 19,624 | 5.53 [5.17, 5.88] | 1252 | 0.12 [0.11, 0.12] | 1086 [1017, 1154] |
| Asian | 2057 | 11.03 [8.94. 13.12] | 255 | 0.11 [0.09, 0.13] | 227 [184, 270] |
| Black | 951 | 17.76 [12.82,22.70] | 206 | 0.12 [0.09, 0.16] | 169 [122, 216] |
| *Female 65+* | | | | | |
| White | 4885 | 3.22 [2.56, 3.88] | 18,315 | 10.44 [8.52, 12.36] | 175 [143, 208] |
| Asian | 130 | 5.60 [0, 12.64] | 852 | 10.51 [1.10, 19.91] | 8 [1, 15] |
| Black | 66 | 10.44 [0, 24.68] | 530 | 7.10 [0.50, 13.69] | 7 [1, 14] |
| *Male <65* | | | | | |
| White | 19,694 | 5.75 [5.32, 6.19] | 2071 | 0.18 [0.17, 0.20] | 1135 [1050, 1221] |
| Asian | 2091 | 13.72 [10.71, 16.74] | 545 | 0.19 [0.15, 0.23] | 287 [224, 351] |
| Black | 888 | 18.43 [13.00, 23.86] | 333 | 0.20 [0.14, 0.26] | 164 [116, 212] |
| *Male 65+* | | | | | |
| White | 3875 | 2.68 [2.08, 3.29] | 21,383 | 17.05 [13.84, 20.26] | 125 [102, 149] |
| Asian | 121 | 4.55 [0, 9.22] | 1361 | 19.79 [3.92, 35.67] | 7 [1, 12] |
| Black | 55 | 7.60 [0, 19.48] | 867 | 17.13 [2.93, 31.33] | 5 [1, 9] |

[a]Estimates of prevalence adjusted for imperfect test sensitivity and specificity, and re-weighted to account for sample design and for variation in response rate (age, sex, ethnicity and deprivation) to be representative of the England population (18+ years).
[b]Infection fatality ratios in (a) were calculated excluding care home residents and based on confirmed COVID-19 death counts obtained from Public Health England[38]. Infection fatality ratios in (b) included deaths in care home residents up to 17 July 2020 obtained from Office for National Statistics[41,42].
[c]Deaths by ethnicity only available including care home residents. The proportions of the population and COVID-19 deaths by ethnicity, age and sex were applied to the current total population of England and the total deaths with COVID-19 on the death certificate up to 17 July 2020. The data sources used are as follows: Population proportions by ethnicity[43]; COVID deaths by ethnicity[38]; 2011 population by ethnicity, age, sex[44]; COVID deaths by ethnicity, age, sex[45].

1.58% (1.51%, 1.65%) if excess rather than COVID-specific deaths are used and care home deaths are included (Supplementary Table 5). There was no difference in estimated IFR for people of Black, Asian and white ethnicities when stratified by age and sex (Table 2).

## Discussion

Overall we estimate a prevalence of SARS-CoV-2 antibody of 6.0% corresponding to 3.4 million adults in England infected by the virus to mid-July 2020. The majority of people who developed antibodies reported symptoms during the peak of the epidemic in

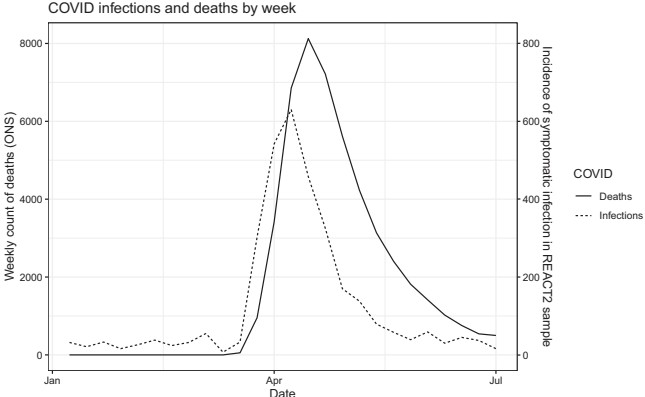

**Fig. 1 Reconstruction of epidemic curve from REACT-2 alongside national reported deaths from COVID-19.** Number of symptomatic infections by week (dotted line; right y axis) based on the date of onset among 3493 antibody-positive participants who reported symptoms in the REACT-2 study, compared with deaths by week in England (solid line; left y axis. Data from Office for National Statistics[46]). Source data are provided with this paper.

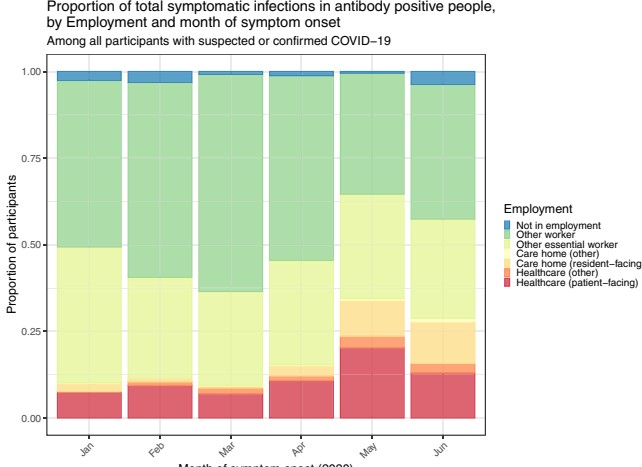

**Fig. 2 Proportion of infections by employment and month of symptom onset.** Proportion of total monthly symptomatic infections based on the date of onset among 3493 antibody-positive participants who reported symptoms in the REACT-2 study, by employment status and month of symptom onset. Source data are provided with this paper.

March and April 2020. As the epidemic took off it became more concentrated in specific groups including Black, Asian and other minority ethnic groups, and in essential workers, particularly those working in health and residential social care. While partially attenuated in the adjusted analyses, the higher risks persisted among these groups and reflect a starkly uneven experience of the COVID-19 epidemic across society.

An unequal burden of COVID-19 morbidity and mortality is emerging from other countries as well as the UK[8,22–25]. Our study has the advantage of including ethnicity data alongside information about employment, deprivation, household size and other potential explanatory variables. This allows a more nuanced exploration of the reasons underlying these unequal outcomes[7]. In the UK context our finding of a higher prevalence of infection, with no apparent difference in IFRs, may explain the observed excess mortality in minority ethnic groups. Therefore there is a need to better understand the occupational, social and

environmental factors that may have led to higher prevalence in these groups[26].

Our estimated IFR of 0.90% is consistent with a recent large study in Spain which reported 0.83–1.07%, lower than the IFR described in Italy (2%), and higher than that reported from a German study (0.38%)[27–29]. In estimating the IFR, we may have underestimated the number of infected individuals (leading to higher estimates of IFR), as a result of weakened or absent antibody response in some people, and waning antibody over time[30]. For the analyses of IFR nationally, we excluded deaths in care home residents since few such residents were included in our community sample. Inclusion of care home residents increased our estimates of IFR, since, like many countries, England experienced high numbers of cases and deaths in care home residents[31]. We included care home residents in our analyses of IFR by ethnicity as data excluding these individulas were not available.

The clinical spectrum of infection is wide, with just under one-third of people with antibodies reporting no symptoms, rising to nearly one half of people over 65 years, as also reported for individuals in long-term care[32]. The national prevalence study in Spain reported that 28.5% or 32.7% were asymptomatic depending on the test[14], similar to our findings overall, although a systematic review of 16 clinical studies puts the figure at 40–50%[33]. The high prevalence of asymptomatic infection means that such cases will be missed by many routine testing campaigns that are based wholly or mainly on symptomatic individuals.

Our finding that current smokers have a lower prevalence of SARS-CoV-2 infection than non-smokers may relect unmeasured confounding, differential adoption of preventive behaviours (given the known associations of COVID-19 severity with smoking-related co-morbidities), or there may be some biological basis. For example, the effect of nicotine on angiotensin converting enzyme 2 (ACE2) receptors, a route of viral entry into cells, has been proposed as a potential mechanism[34].

Our study has a number of limitatons. As in almost all population surveys, our study had unequal participation, with lower response among people from minority ethnic groups and in more deprived areas. We re-weighted the sample to account for such differential response, although this may not have overcome unknown participation biases. An important limitation was the exclusion of children for regulatory reasons as the tests were approved for research use in adults only. Furthermore, our sampling approach only allowed for one individual per household to take part in the study thus limiting our ability to explore the impact of household transmission on associations seen with other covariates. However, we did control for household size in our regression analysis to account for this. Numbers were too small to report the ethnic breakdown of antibody prevalence according to more detailed categories, as such important differences between ethnic sub-groups with respect to occupation, deprivation, and region may not have been fully captured. We used self-administered home LFIA tests as opposed to "gold standard" laboratory tests on a blood draw. However, this followed an extensive evaluation of the selected LFIA whch showed it to have acceptable performance (sensitivity and specificity) in comparison with confirmatory laboratory tests[19]. We also took steps to measure and improve usability, including ability to perform and read an LFIA test at home, through public involvement and evaluation in a national study of 14,000 people[20].

Use of the LFIA enabled us to obtain antibody tests on large numbers of people over an 18-day period, without the need for laboratory testing or health care personnel. Antibodies were strongly associated with clinical history of confirmed or suspected COVID-19, providing face validity. Although there was a theoretical potential for reporting bias as respondents were not

blinded to their test results, there was high concordance of self-reported with clinician-read results from the uploaded photographs[20]. Our results closely tracked other indicators of the epidemic curve. We believe that use of home-based self-tests is a sustainable model for community-based prevalence studies in other populations, avoiding the biases of surveillance that relies solely on self-referral for testing. Continued scrutiny of antibody response by clinical features, and persistence of antibodies over time, will be needed for ongoing surveillance, as waning antibodies mean that prevalence estimates may not fully capture cumulative exposure over time.

In conclusion, our finding of substantial inequalities in prevalence of SARS-CoV-2 infection by ethnicity runs counter to suggestions that the increased risk of hospitalisation and mortality from COVID-19 among minority ethnic groups is due predominantly to comorbidities or other biological factors. Work with at risk communities is urgently needed to identify appropriate interventions to reduce health inequalities related to risk of SARS-CoV-2 infection.

## Methods

The REal-time Assessment of Community Transmission-2 (REACT-2) programme is evaluating community prevalence of SARS-CoV-2 infection in England. We obtained a random population sample of adults in England, using the National Health Service (NHS) patient list, which includes name, address, age and sex of everyone registered with a general practitioner (almost the entire population). Personalised invitations were sent via post to 315,000 individuals aged 18 years and above to achieve similar numbers in each of 315 lower-tier local authority areas (LTLAs). Participants registered via an online portal or by telephone with registration closed after ~120,000 people had signed up. To attain approximately the same number of registrations per LTLA, the number of invitations sent varied based on the LTLA response profile achieved when conducting similar population surveys in England[35].

Those registered were sent a test kit, including a self-administered point-of-care LFIA test and instructions by post, with link to an on-line video. The questionnaires are available at the study website: https://www.imperial.ac.uk/medicine/research-and-impact/groups/react-study/react-2-study-materials/. The LFIA (Fortress Diagnostics, Northern Ireland) was selected following evaluation of performance characteristics (sensitivity and specificity) against pre-defined criteria for detection of IgG[19], and extensive public involvement and user testing[20]. The LFIA uses the coronavirus structural spike (S) protein as the target antigen for the antibody-based detection of SARS-CoV-2. Compared to results from at least one of two in house ELISAs, sensitivity and specificity of finger-prick blood (self-read) were 84.4% (70.5%, 93.5%) in RT-PCR confirmed cases and 98.6% (97.1%, 99.4%) in 500 pre-pandemic sera[19]. The in-house ELISAs used in that evaluation of the LFIA were the spike protein ELISA (S-ELISA) and a hybrid spike protein receptor-binding domain double antigen-bridging assay (hybrid DABA)[19]. Samples for sensitivity testing were collected from adult NHS workers, who had previously tested positive for SARS- CoV-2 by PCR, but not hospitalised and were at least 21 days from the onset of symptoms[19].

Participants completed a short registration questionnaire (online/telephone) and a further survey upon completion of their self-test. This included information on demographics, household composition, recent symptoms and an uploaded photograph of the result. A validation study of the photographs showed substantial concordance between participant- and clinician-interpreted results in over 500 tests (kappa: 0.89, 95% CI: 0.88–0.92)[20].

Prevalence was calculated as the proportion of individuals with a positive IgG result, adjusted for test performance using:

$$p = (q + \text{specificity} - 1)/(\text{sensitivity} + \text{specificity} - 1) \qquad (1)$$

where $p$ is the adjusted proportion positive, $q$ is the observed proportion positive[17]. Prevalence estimates at national level were weighted for age, sex, region, ethnicity and deprivation to account for the geographic sample design and for variation in response rates, so as to be representative of the population (18+ years) of England. Details of the weighting approach used and the sample population profile are in the Supplementary Information. Logistic regression models were adjusted for age, sex and region, and additionally for ethnicity, deprivation, household size and occupation. We used complete case analysis without imputation.

Regions are the highest tier of sub-national division in England and are predominantly used for statistical and some administrative purposes, London being the most dense and urban region, and the South West the least dense and most rural (further details in Supplementary Information). Index of Multiple Deprivation 2019 (IMD) was used as a measure of relative deprivation, based on seven domains at a small local area level across England (income, employment, education, health, crime, barriers to housing and services, and living environment)[36].

We estimated total number of SARS-CoV-2 infections since start of the epidemic until mid-July 2020 by multiplying the antibody prevalence, adjusted for test characteristics and re-weighted for sampling, by mid-year population size at ages 18+ years in England[37]. To correct for survival bias we added to the seropositive population the deaths that mentioned COVID-19 on the death certificate during this period. Office for National Statistics (ONS) COVID-19 deaths registration data used includes deaths where COVID-19 was recorded as a cause of death on the death certificate, whether or not there was a laboratory-confirmed test and, at the time, irrespective of the interval from date of testing positive for those who were tested[38]. We then estimated the IFR, dividing the total number of COVID-19 deaths excluding care home residents[14]. We obtained an overall IFR estimate and estimates stratified by age and sex[15]. We calculated the IFR without care home deaths since we did not have sufficient numbers of care home residents in our study to be able to get an accurate estimate of prevalence of infection in this population. Early data suggested that the rate of infection in care homes was higher than in the general population[39], and therefore including care home deaths would overestimate the IFR. We present separate IFR estimates by ethnicity because of the lack of availability of data on COVID deaths disaggregated by both ethnicity and care home residency, therefore we could not exclude COVID-19 deaths in care home residents from each ethnic group. Confidence bounds were obtained using the Delta method. As a sensitivity analysis we calculated IFR and total infections including care home residents, with all-cause excess deaths and stratified by age and sex. ONS excess mortality is defined as the number of deaths in 2020 which are above the number expected based on mortality rates in earlier years[38]. We obtained research ethics approval from the South Central-Berkshire B Research Ethics Committee (IRAS ID: 283787), and MHRA approval for use of the LFIA for research purposes only, and participants provided informed consent.

Data were analysed using the statistical package R version 4.0.0[40].

**Reporting summary.** Further information on research design is available in the Nature Research Reporting Summary linked to this article.

## Data availability

Source data are provided with this paper.

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

## Acknowledgements

This work was funded by the Department of Health and Social Care in England. The content of this manuscript and decision to submit for publication were the responsibility of the authors and the funders had no role in these decisions. H.W. is a NIHR Senior Investigator and acknowledges support from NIHR Biomedical Research Centre of Imperial College NHS Trust, NIHR School of Public Health Research, NIHR Applied Research Collaborative North West London, Wellcome Trust (UNS32973). G.C. is supported by an NIHR Professorship. W.B. is the Action Medical Research Professor, A.D. is an NIHR senior investigator and D.A. is an Emeritus NIHR Senior Investigator. S.R. acknowledges support from MRC Centre for Global Infectious Disease Analysis, National Institute for Health Research (NIHR) Health Protection Research Unit (HPRU), Wellcome Trust (200861/Z/16/Z, 200187/Z/15/Z), and Centres for Disease Control and Prevention (US, U01CK0005-01-02). P.E is Director of the MRC Centre for Environment and Health (MR/L01341X/1, MR/S019669/1). P.E. acknowledges support from the NIHR Imperial Biomedical Research Centre and the NIHR HPRUs in Environmental Exposures and Health and Chemical and Radiation Threats and Hazards, the British Heart Foundation Centre for Research Excellence at Imperial College London (RE/18/4/34215) and the UK Dementia Research Institute at Imperial (MC_PC_17114). We thank key collaborators on this work—Ipsos MORI: Stephen Finlay, John Kennedy, Kevin Pickering, Duncan Peskett, Sam Clemens and Kelly Beaver; Institute of Global Health Innovation at Imperial College London: Gianluca Fontana, Dr. Hutan Ashrafian, Sutha Satkunarajah and Lenny Naar; the Imperial Patient Experience Research Centre and the REACT Public Advisory Panel; NHS Digital for access to the NHS Register.

## Author contributions

H.W. and P.E. designed the study and drafted the manuscript. M.W., K.E.C.A., J.E., C.A., L.O., R.R. conducted the analyses. H.W., C.A., G.C., D.A., C.A.D., W.B., A.D., G.C., S.R., P.E., provided study oversight. A.D. and P.E. obtained funding. All authors have reviewed and approved the final manuscript.

## Competing interests

The authors declare no competing interests.
