## [Peer Review File · Nature Communications]

REVIEWER COMMENTS

Reviewer #1 (Remarks to the Author):

Ward et al. present a very elegant analysis of more than 100,000 people self-tested for SARS-CoV-2 antibodies. They provide valuable additional information on inequalities in SARS-COV-2 infection. The use of community-wide estimates are indeed important. The stratification by non-essential and essential workers, health care and non-health care workers, and ethnicity is appreciated.

Major concerns:

1) The authors acknowledged the lack of information on household infections. This is actually a major concern because household infection can confound seriously the reported associations. There is a growing body of evidence suggesting that the risk of SARS-COV-2 infection within households is major. I strongly suggest that further address this concerns (e.g., restrict their analysis to household with a maximum of members and check whether the associations [care workers, ethnicity, et] persist.

2) Major limitation is the lack of information on education and income, region-specific data should at least be considered

3) Household information is appreciated but can be misleading without considering whether it is mostly kids or without information in overcrowding (which may drive infection rate more than just the number of subjects in a house)

4) To interpret the results correctly, the readers should have a sense of the public health measures implemented for each period that could have influenced the risk of infection (lock down, restriction, visits, etc) and the assess to material (mask, etc) for healthcare providers

5) I appreciate the methodology published in the Study protocol (Wellcome Open Research) but I suggest that more information on the methodology should be available (some footnote mention more details in methods but details are actually not reported) in the manuscript itself including on:

- Weighted method and re-weighting method
- Handling of missing data and the rationale of using case analysis without imputation. Suggest to run analysis with imputation and assess the results.
- Death certification collection
- Participation rate response by means of contact
- Deprivation definition

Additional comments:

1) It is difficult for international readers of Nature Communication to figure out the difference between region characteristics (North East vs North West). Authors should characterize these regions differently (urban sub urban rural services etc)

2) In table 1 what is the rationale of presenting the results by sex and age only for IRF a) and by ethnicity for b)?

3) When reporting logistic regression by covariates, provide information of lockdown measure for the different age categories, region (see comment above), overcrowding measure instead of household size.

4) Not clear what the last covariate of the figure 1 is

5) Figure 2 panel c is not really contributive by reporting name of regions without their characteristics

Reviewer #2 (Remarks to the Author):

The title is confusing – this is an important seroprevalence study and not a study exploring measures of deprivation/ethnicity and their relation to SARS2 infection (which would be an interesting analysis in its own right). I therefore would suggest the authors change the title to something more accurate such as 'SARS-CoV-2 antibody prevalence in England following the first peak of the pandemic'.

The overall results are not surprising. The ONS survey produced similar seroprevalence data (ie a seroprevalence of 6% across England). Higher seroprevalance in healthcare workers and in Black and Asian communities has also been reported elsewhere. It is unclear to me whether the study can disentangle race, deprivation and the proportion of people-facing healthcare workers of Black or Asian origin causing increased transmission in those communities. I believe that a public health statistician (not a clinician or scientist) would have to review the paper to determine if the study has the statistic power to do so. The title and summary seem to suggest that it has an answer to these important questions.

As the study is measuring antibody levels in sera, I would suggest that the authors state which antigen is used in the Fortress assay, and also the antigen used in the two in-house ELISAs to determine the sensitivity of the Fortress assay. It is surprising that this basic information has not been included.

The type of positive samples used to determine the validity of the assay would also be of interest – were they contract-traced asymptomatic individuals or convalescent samples? The latter of these would be expected to have higher antibody levels and thus will affect the Fortress assay sensitivity measurements.

Other than these suggestions and highlighted issues with the study, I think the study is robust and the authors should be congratulated on successfully undertaking such a large-scale analysis.

Reviewer #3 (Remarks to the Author):

Many thanks for giving me the opportunity to review this manuscript from the REACT-2 study team. This study is hugely important and it is wonderful that the results are being reported in such a timely manner, particularly as infection levels are rising again across the UK and Europe. The study fills a key knowledge gap around community prevalence of COVID-19 in an unselected population based sample and has important strengths around being able to report on ethnicity, deprivation, occupation, and region. The manuscript is clearly written and reports on the key outcomes clearly. I provide some suggestions below for clarifying some aspects of the study design and overall reporting.

1. Line 203: Can this sentence be made clearer: Personalized invitations were sent to 315,000 204 individuals aged 18 years and above to achieve similar numbers in each of 315 lower-tier 205 local authority areas. Does this mean that you sent invitations to an equal number of people from each local authority area?

2. Line 206: It would be useful for the authors to clarify the extent to which their included population was representative of the general population as stated. You state that registration was closed after 120,000 signed up, were checks for population representativeness with respect to age,

ethnicity, and occupation undertaken at this stage, or was this representativeness only achieved in the re-weighting of the test results?

3. Were you able to compare any characteristics between people who did and didn't register to participate in the study? I.e. was interest in participation related to socio-demographic characteristics, occupation, region?

4. Line 223: Clarify briefly where the population weights were derived from (The NHS patient lists, census data, other?)

5. Line 231: How was death certificate obtained? Did participants consent to linkage of health or ONS data? I could not find this information in the protocol pre-print on Wellcome Open Research or in the extended data. Similarly, was information on demographic/exposure characteristics and care home residency derived from the questionnaire or another source? I could not see this information clarified in the text here, nor could I find the actual questionnaire as supplementary material as part of either this submission or the Wellcome Open protocol.

6. You give a reference for the exclusion of care home residents from the study population, but an additional sentence clarifying the justification for this would be useful.

7. Can the authors clarify some of the categories used in the tables reporting prevalence and odds ratios. For example, for employment 'Other Worker' is the reference category. There is no explanation anywhere of what this means. I think it is meant to be people who don't work in people facing/frontline roles, but this isn't made clear anywhere.

8. Furthermore for the category 'contact with a case' - did the respondents have the opportunity to answer that they don't know, and if so, was prevalence/odds calculated for this category?

9. Standard BMI categories are used in the tables, since ethnic differences are a key question being addressed here, I would recommend using ethnic specific cut-offs for the categories- at least as a sensitivity analysis. Lower cut-offs for BMI categories for people of south Asian ethnicity are now part of UK NICE guidelines and these may better reflect differences in risk associated with BMI in a multi-ethnic population.

11. Did the authors have enough individuals to report the ethnic breakdown according to more detailed categories such as African, Caribbean, Bangladeshi, Pakistani, Indian etc. There are established important differences between these ethnic sub-groups with respect to occupation, deprivation, and region which this study is well placed to shed light on and will also serve a very important public health and policy planning need. If this is not possible, some acknowledgement that high level categories may not fully capture important differences by ethnic group should be made.

12. The finding that inequalities in mortality may be related to infection risks rather than biological risk factors of access to care is key. However, it is important to clarify whether or not these data can tell us anything about the role of biological factors on increased risk of infection amongst those exposed to the virus?

Minor comments

Line 71-73: It would be useful to report the number(percent) consistently for each reported population group i.e. number who completed the survey and antibody test, number who were IgG positive and negative.

In tables 1 & 3 I would perhaps swap the columns for total antibody positive and total tests, to show the 'denominator' first and outcome of interest second - but this is just a suggestion.

Response to reviewers

Reviewer #1 (Remarks to the Author):

Ward et al. present a very elegant analysis of more than 100,000 people self-tested for SARS-CoV-2 antibodies. They provide valuable additional information on inequalities in SARS-CoV-2 infection. The use of community-wide estimates are indeed important. The stratification by non-essential and essential workers, health care and non-health care workers, and ethnicity is appreciated.

Response: thank you

Major concerns:

- 1) The authors acknowledged the lack of information on household infections. This is actually a major concern because household infection can confound seriously the reported associations. There is a growing body of evidence suggesting that the risk of SARS-CoV-2 infection within households is major. I strongly suggest that further address this concerns (e.g., restrict their analysis to household with a maximum of members and check whether the associations [care workers, ethnicity, et] persist.

Response: We do not have multiple observations within households. We control for household size in the jointly adjusted regression analysis (Figure 1 and Extended data table 2) and the associations with care workers and ethnicity etc. persist. We have clarified this in the DISCUSSION by adding, line 185-190

"Furthermore, our sampling approach only allowed for one individual per household to take part in the study thus limiting our ability to explore the impact of household transmission on associations seen with other covariates. However, we did control for household size in our regression analysis to account for this."

- 2) Major limitation is the lack of information on education and income, region-specific data should at least be considered

Response: We include index of multiple deprivation (IMD) in our analysis which includes income and education at the area level. IMD is an area level measure (sub-regional, relating to 1000 to 3000 people) with seven domains that include income, employment, education, health/disability, crime, housing, environment. We have added a sentence to the methods (line 259-262) to explain, and a citation. However, in response to your query we explored the association with individual educational level (highest qualification) and household income. Prevalence was lowest in those with education to degree-level or above 3.9% [3.4-4.4], and highest in those with no qualifications 5.7% [5.4-6.0], now added to Extended Data table 3.

Interestingly, prevalence was higher in those with higher household incomes. We have also added this to Figure 1 and Extended Data Table 3. This finding contrasts with the group level observation of a higher prevalence in more deprived areas. The association of prevalence with higher household income reduced slightly after controlling for age, sex and Region, but was not altered by further adjustments including ethnicity, employment and household size (Extended data table 2). We also did a specific exploration of the

relationship with age, and show (Extended data table 5) that the effect was strongest in the younger age groups. This may relate to early multiple introductions of the virus into the country by people returning from e.g. ski holidays in Europe. We have added this to the MAIN text (line 110): "In contrast, higher household income was associated with increased prevalence of antibodies (Figure 1, Extended data Table 2), which was greater in younger age groups (Extended data Table 5)."

- 3) Household information is appreciated but can be misleading without considering whether it is mostly kids or without information in overcrowding (which may drive infection rate more than just the number of subjects in a house)

Response: *We have now added the impact of number of children in a household on prevalence and in relation to household size. We find that having any children in the household is associated with a small increase in prevalence compared to no children (but more children did not further increase the risk); however, in a regression model (now included in extended data table 2) the association disappears in adjusted models including age, sex and region. This suggests that the increased risk from larger households may be due to increased numbers of adults primarily. We have included a comment on this in the results.*

- 4) To interpret the results correctly, the readers should have a sense of the public health measures implemented for each period that could have influenced the risk of infection (lock down, restriction, visits, etc) and the assess to material (mask, etc) for healthcare providers.

Response: *We have added a description of this into the text. Lines 51-55 "As part of the UK Government's response to controlling the spread of the virus, on March 23 it announced a national lockdown that prohibited all but essential activities. The UK then came out of lockdown from mid-May as restrictions were gradually eased as more business were allowed to reopen and members of the public were encouraged to use face coverings in situations when social distancing could not be maintained."*

- 5) I appreciate the methodology published in the Study protocol (Wellcome Open Research) but I suggest that more information on the methodology should be available (some footnote mention more details in methods but details are actually not reported) in the manuscript itself including on:

- a. Weighted method and re-weighting method:

Response: *We have added a supplementary section in the appendix on weighting (S1)*

- b. Handling of missing data and the rationale of using case analysis without imputation. Suggest to run analysis with imputation and assess the results.

Response: *We have provided detail on missing data by variable in the supplementary material. With the exception of BMI there are few missing data (see supplementary methods figure S2) and since BMI is not included in the regression models we do not think imputation is indicated.*

c. Death certification collection

Response: *We did not collect death certification data ourselves but used published aggregate data on deaths which is defined in various ways. We have now clarified this in the methods with the following addition (299-303): "METHODS: Office for National Statistics (ONS) COVID-19 deaths registration data used include deaths where COVID-19 was recorded as a cause of death on the death certificate, whether or not there was a laboratory-confirmed test and, at the time, irrespective of the interval from date of testing positive for those who were tested. ONS excess mortality is defined as the number of deaths in 2020 which are above the number expected based on mortality rates in earlier years."*

d. Participation rate response by means of contact

Response: *Personalized invitations to participate in the study were all sent via post. Therefore, we do not have separate response rates by "means of contact" as all potential participants were contacted by letter in the post. Details of participation rates are provided in Supplementary methods table S2*

e. Deprivation definition

Response: *We have added a short explanation and citation in methods (line 291-294): "Index of Multiple Deprivation 2019 (IMD) was used as a measure of relative deprivation, based on seven domains at a small local area level across England (income, employment, education, health, crime, barriers to housing and services, and living environment)."*

Other comments

- 1) It is difficult for international readers of Nature Communication to figure out the difference between region characteristics (North East vs North West). Authors should characterize these regions differently (urban sub urban rural services etc)

Response: *We have provided a short description in the methods and more detail in the supplementary material (lines 288-291): METHODS: Regions are the highest tier of sub-national division in England and are predominantly used for statistical and some administrative purposes, London being the most dense and urban region, and the South West the least dense and most rural (further details in Supplementary material section 4, table S3, figure S1).*

- 2) In table 1 what is the rationale of presenting the results by sex and age only for IRF a) and by ethnicity for b)?

Response *This was done because of lack of availability of data for the appropriate strata. We have clarified this in METHODS (lines 309-317): We present separate IFR estimates by ethnicity in table 1(b) because of the lack of availability of data on COVID deaths disaggregated by both ethnicity and care home residency; therefore we could not exclude COVID-19 deaths in care home residents from each ethnic group.*

- 3) When reporting logistic regression by covariates, provide information of lockdown measure for the different age categories, region (see comment above), overcrowding measure instead of household size.

Response: *There were no differences in lockdown measures between March and June for age categories or regions; overcrowding is a component part of IMD, and therefore controlled for in part.*

- 4) Not clear what the last covariate of the figure 1 is

Response: *apologies, this has been relabelled.*

- 5) Figure 2 panel c is not really contributive by reporting name of regions without their characteristics

Response: *We agree and we have removed panels C and D which were both unclear.*

Reviewer #2 (Remarks to the Author)

The title is confusing – this is an important seroprevalence study and not a study exploring measures of deprivation/ethnicity and their relation to SARS2 infection (which would be an interesting analysis in its own right). I therefore would suggest the authors change the title to something more accurate such as ‘SARS-CoV-2 antibody prevalence in England following the first peak of the pandemic’.

Response: *thank you for this suggestion, we agree and have made the change to: SARS-CoV-2 antibody prevalence in England following the first peak of the pandemic: the REACT2 study*

The overall results are not surprising. The ONS survey produced similar seroprevalence data (ie a seroprevalence of 6% across England). Higher seroprevalance in healthcare workers and in Black and Asian communities has also been reported elsewhere. It is unclear to me whether the study can disentangle race, deprivation and the proportion of people-facing healthcare workers of Black or Asian origin causing increased transmission in those communities. I believe that a public health statistician (not a clinician or scientist) would have to review the paper to determine if the study has the statistic power to do so. The title and summary seem to suggest that it has an answer to these important questions.

Response: *The study was powered to look at variation in prevalence at small area level, and sample size calculations are available in the published protocol. While high level findings are indeed similar to the findings from ONS, our study is larger as reflected in the narrower confidence intervals for the weighted estimate for England: REACT2: 6.0% (95% C 5.8-6.1), ONS June 5.9% (4.5-7.6). With this size of study we have been able to explore the factors mentioned by the reviewer (race, deprivation, employment) in our regression models which are presented with confidence intervals. As an example, logistic regression*

found a significant association between seropositive status and minority ethnic groups after adjusting for deprivation and employment type.

As the study is measuring antibody levels in sera, I would suggest that the authors state which antigen is used in the Fortress assay, and also the antigen used in the two in-house ELISAs to determine the sensitivity of the Fortress assay. It is surprising that this basic information has not been included. The type of positive samples used to determine the validity of the assay would also be of interest – were they contract-traced asymptomatic individuals or convalescent samples? The latter of these would be expected to have higher antibody levels and thus will affect the Fortress assay sensitivity measurements.

Response: *This information has previously been published in the cited paper (Flower B et al, 2020), but to clarify we have added the following to the METHODS (263-271): The LFIA uses the coronavirus structural S protein as the target antigen for the antibody-based detection of SARS-CoV-2. The in-house ELISAs used in that evaluation of the LFIA were the spike protein ELISA (S- ELISA) and a hybrid spike protein receptor binding domain double antigen bridging assay (hybrid DABA). Samples for sensitivity testing were collected from adult NHS workers, who had previously tested positive for SARS-CoV-2 by PCR, but not hospitalised and were at least 21 days from the onset of symptoms.*

Other than these suggestions and highlighted issues with the study, I think the study is robust and the authors should be congratulated on successfully undertaking such a large-scale analysis.

Response: *thank you*

Reviewer #3 (Remarks to the Author):

Many thanks for giving me the opportunity to review this manuscript from the REACT-2 study team. This study is hugely important and it is wonderful that the results are being reported in such a timely manner, particularly as infection levels are rising again across the UK and Europe. The study fills a key knowledge gap around community prevalence of COVID-19 in an unselected population based sample and has important strengths around being able to report on ethnicity, deprivation, occupation, and region. The manuscript is clearly written and reports on the key outcomes clearly. I provide some suggestions below for clarifying some aspects of the study design and overall reporting.

Response: *thank you*

1. Line 203: Can this sentence be made clearer: Personalized invitations were sent to 315,000 204 individuals aged 18 years and above to achieve similar numbers in each of 315 lower-tier 205 local authority areas. Does this mean that you sent invitations to an equal number of people from each local authority area?

Response: The number of invitations per LA varied according to expected response rate, and were set in each LA to achieve approximately the same number of registrations per LA. This was based on LA response profile to invitations to the first REACT1 round. We have clarified in the text (lines 253-256) and added a citation: METHODS: To attain approximately the same number of registrations per LTLA, the number of invitations sent varied based on the LTLA response profile achieved when conducting similar population surveys in England.

2. Line 206: It would be useful for the authors to clarify the extent to which their included population was representative of the general population as stated. You state that registration was closed after 120,000 signed up, were checks for population representativeness with respect to age, ethnicity, and occupation undertaken at this stage, or was this representativeness only achieved in the re-weighting of the test results?

Response to 2 and 3: The registration profile was monitored in relation to the attributes that were included in the sample frame: sex, age groups, and region, but could not be for ethnicity and occupation as they are not included in the sample frame. Plans to balance the registration profile through targeted reminders were not carried out given the rapid rate of response (i.e. the speed with which people registered). Once the sample was complete it was compared to the base population and, as in almost all population surveys, our study had some unequal participation, with lower response among younger people, men, minority ethnic groups and in more deprived areas. We re-weighted the sample to account for differential response (details now in supplementary material section 1). Details on the sample profile invited and achieved are provided in Supplementary material section 2 and table S1, and we have referred to these in the methods section.

3. Were you able to compare any characteristics between people who did and didn't register to participate in the study? I.e. was interest in participation related to socio-demographic characteristics, occupation, region?

Response: yes, see response above

4. Line 223: Clarify briefly where the population weights were derived from (The NHS patient lists, census data, other?)

Response: see response above and detail in supplementary material section 1.

5. Line 231: How was death certificate obtained? Did participants consent to linkage of health or ONS data? I could not find this information in the protocol pre-print on Wellcome Open Research or in the extended data. Similarly, was information on demographic/exposure characteristics and care home residency derived from the questionnaire or another source? I could not see this information clarified in the text here, nor could I find the actual questionnaire as supplementary material as part of either this submission or the Wellcome Open protocol.

Response: We did not link individual health records, death certification or ONS data to survey responses. Summary data were obtained from routine reports available online. Demographic, behavioural and residency data were captured from the questionnaire which is available online <https://www.imperial.ac.uk/medicine/research-and-impact/groups/react-study/react-2-study-materials/> (we have added a link to this in the methods section). To clarify on death data we have added the following to the methods (299-303): “METHODS: Office for National Statistics (ONS) COVID-19 deaths registration data used include deaths where COVID-19 was recorded as a cause of death on the death certificate, whether or not there was a laboratory-confirmed test and, at the time, irrespective of the interval from date of testing positive for those who were tested. ONS excess mortality is defined as the number of deaths in 2020 which are above the number expected based on mortality rates in earlier years.”

6. You give a reference for the exclusion of care home residents from the study population, but an additional sentence clarifying the justification for this would be useful.

Response: We have clarified the methods line METHODS: “We calculated the IFR without care home deaths since we did not have sufficient numbers of care home residents in our study to be able to get an accurate estimate of prevalence of infection in this population. Early data suggested that the rate of infection in care homes was higher than in the general population, and therefore including care home deaths would overestimate the IFR in the community.”

7. Can the authors clarify some of the categories used in the tables reporting prevalence and odds ratios. For example, for employment ‘Other Worker’ is the reference category. There is no explanation anywhere of what this means. I think it is meant to be people who don’t work in people facing/frontline roles, but this isn’t made clear anywhere.

Response: Apologies. We have added detail of the categories to the legends of the tables, including Added to Extended Data Table 2: †Essential worker: List of essential workers and those prioritised for testing (England only) as defined by the UK Government available at <https://www.gov.uk/guidance/coronavirus-covid-19-getting-tested#essential-workers> §Other worker: Worker not working in health or social care or on the UK Government list of essential workers.

8. Furthermore for the category ‘contact with a case’ - did the respondents have the opportunity to answer that they don’t know, and if so, was prevalence/odds calculated for this category?

Response: The categories were: (1) Yes, I have been in contact with a confirmed/tested COVID-19 case; (2) Yes, I have been in contact with a suspected COVID-19 case; (3) No, not to my knowledge. We have amended the Extended Data Table 3 to clarify

9. Standard BMI categories are used in the tables, since ethnic differences are a key question being addressed here, I would recommend using ethnic specific cut-offs for the categories- at least as a sensitivity analysis. Lower cut-offs for BMI categories for people of

south Asian ethnicity are now part of UK NICE guidelines and these may better reflect differences in risk associated with BMI in a multi-ethnic population.

Response: *We carried out a sensitivity analysis using the lower cut-offs for BMI suggested, but this made little difference to the results (data not shown). We have added a note to this effect (line 122-124): The association with overweight and obesity was not altered in a sensitivity analysis using different cut-off rates for people of South Asian ethnicity (data not shown).*

- 10.** Did the authors have enough individuals to report the ethnic breakdown according to more detailed categories such as African, Caribbean, Bangladeshi, Pakistani, Indian etc. There are established important differences between these ethnic sub-groups with respect to occupation, deprivation, and region which this study is well placed to shed light on and will also serve a very important public health and policy planning need. If this is not possible, some acknowledgement that high level categories may not fully capture important differences by ethnic group should be made.

Response: *We agree that the ethnicity categories are broad, and we have added the prevalence by more detailed categories Extended Data Table 6, and referred to this in lines 94-97):*

- 11.** The finding that inequalities in mortality may be related to infection risks rather than biological risk factors of access to care is key. However, is it important to clarify whether or not these data can tell us anything about the role of biological factors on increased risk of infection amongst those exposed to the virus?

Response: *We agree and have amended the discussion to clarify (line 173-177), which now reads in DISCUSSION: "In the UK context, our finding of a higher prevalence of infection, with no apparent difference in infection fatality ratios, may explain the observed excess mortality in minority ethnic groups. Therefore there is a need to better understand the occupational, social and environmental factors that may have led to the higher prevalence in these groups."*

Minor comments

Line 71-73: It would be useful to report the number(percent) consistently for each reported population group ie/ number who completed the survey and antibody test, number who were IgG positive and negative.

Response: *these have now been included (line 79-82)*

In tables 1 & 3 I would perhaps swap the columns for total antibody positive and total tests, to show the 'denominator' first and outcome of interest second - but this is just a suggestion.

Response: *this has been done, thank you for the suggestion.*

REVIEWERS' COMMENTS

Reviewer #3 (Remarks to the Author):

I thank the authors for their considered responses to my comments. I am happy to recommend the revised manuscript for acceptance.